Accepted at the ICLR 2024 Workshop on AI4Differential Equations In Science

# APPLICATIONS OF FOURIER NEURAL OPERATORS IN THE IFMIF-DONES ACCELERATOR

**Guillermo Rodríguez Llorente, Galo Gallardo Romero & Roberto Gómez-Espinosa Martín** [*]
Department of Artificial Intelligence
HI Iberia
Juan Hurtado de Mendoza, 14, 28036 Madrid, Spain
`{grodriguez,ggallardo,robertogemartin}@hi-iberia.es`

## ABSTRACT

In this work, Fourier Neural Operators are employed to improve control and optimization of an experimental module of the IFMIF-DONES linear accelerator, otherwise hindered by its simulations high complexity. The models are trained to predict beam envelopes along the lattice's longitudinal axis, considering variations in quadrupole strengths and particle injections. They serve three purposes: enabling fast inference of beam envelopes, creating an environment for training a Deep Reinforcement Learning agent responsible for shaping the beam, and developing an optimizer for identifying optimal accelerator parameters. The resulting models offer significantly faster predictions (up to 3 orders of magnitude) compared to traditional simulators, with maximum percentage errors below 2 %. This accelerated simulation capability makes it feasible to train control agents, since the time per step taken is reduced from 3s to $4 \times 10^{-3}$s. Additionally, Stochastic Gradient Descent was applied to optimize one of the models itself, determining the best parameters for a given target and thus solving the inverse problem within seconds. These results demonstrate the synergy and potential of these Deep Learning models, offering promising pathways to advance control and optimization strategies in the IFMIF-DONES accelerator and in other complex scientific facilities.

## 1 INTRODUCTION

One of the foremost challenges in future nuclear fusion power plants is studying the effects of neutron irradiation on reactor materials. In addressing this issue, the IFMIF-DONES facility aims to build a linear accelerator in order to generate a neutron spectrum similar to that of a fusion reaction by the collision of deuterons with a lithium circuit (Bernardi et al., 2022). This study, conducted within the DONES-FLUX project, explores the joint application of Deep Learning Surrogate Models (DLSMs) and Deep Reinforcement Learning (DRL) in order to improve the optimization and control of the facility.

DLSMs are models of physical, chemical, or biological processes based on Artificial Intelligence (AI), that enable rapid and accurate high-fidelity simulations. This technique provides a method to accelerate complex simulations, governed by partial differential equations, with data (Hao et al., 2023). On the other hand, DRL is an AI paradigm that is applied to solve sequential decision-making problems (Graesser & Keng, 2019). In this approach, an agent interacts with an environment (usually simulated) that provides feedback in the form of rewards or penalties based on its actions. DRL-based controllers can improve the development and efficiency of traditional systems, allowing agents to achieve goals set by experts instead of constantly making manual adjustments (Degrave et al., 2022). Training DRL agents using traditional simulators becomes impractical due to the substantial time required to process each step. In this context, the use of DLSMs helps expedite execution times, enabling the online learning of such agents.

In this work, we train two Fourier Neural Operator (FNO) (Li et al., 2020) models with data from the OPAL simulator (Adelmann et al., 2019). The selection of this architecture is motivated by the fact

---

[*]https://www.hi-iberia.es

that it is purely data driven, since there is a lack of comprehensive knowledge regarding the differential equations that govern the transfer of beam bunches across the accelerator. Moreover, the FNO is discretization invariant which enables multi-resolution approaches, an advantage compared with a neural network Azizzadenesheli et al. (2024). The simulator makes use of the Monte Carlo method for the interactions of particles along the lattice, which is stochastic. This is a computationally expensive process which incentives the development of model capable of predicting the macroscopic distributions and envelopes of the particles. The distribution of particles along the accelerator is a two-dimensional Gaussian Wiedemann (2015), for this reason for our purposes of training DRL agents, predicting only the beam envelope (standard deviation along the two transversal axes) can be seen as a form of feature engineering, in which we are reducing an observation space of 20000 particles to a two-dimensional space, otherwise untraceable.

The trained models are able to predict the deuteron envelopes along the longitude of an experimental module of the linear accelerator, displayed in the lower part of Fig. 1. These models find three different applications: fast prediction of beam envelopes, training of a DRL agent that controls the geometry of the beam at the end of the lattice, and building an optimizer to find the best parameters for a given configuration. The problem can be set as design problem if the geometry of the beam is static or as a real time control problem if the geometry is dynamic (for instance, the currents of the quadrupoles can be controlled). For the second option, the DRL agent can be used to optimize the design of the lattice and to correct the beam geometry in real time, as the footprint at the target needs to have a specific shape for the reaction. For details on FNOs and DRL, refer to Section 2. Section 3 presents the methodology and obtained results, while the discussion is encapsulated in Section 4.

## 2 MODELS

Neural operators (NOs) represent an emerging branch of DLSMs, extending the architecture of neural networks for the approximation of operators capable of mapping between infinite-dimensional function spaces (Kovachki et al., 2023). This generalization becomes particularly relevant for solving systems of partial differential equations, which can ultimately represent physical phenomena. The main architecture of these models takes data from input functions $a(x)$ to predict output functions $u(x)$, as can be seen in the upper part of Fig. 1. The layer $P$ maps the input function to its first hidden representation, which is then passed through layers of kernels with non-linear activation functions, and finally taken to the output representation with the layer $Q$. The FNO model employs the Fourier transform for computing the kernels (Li et al., 2020).

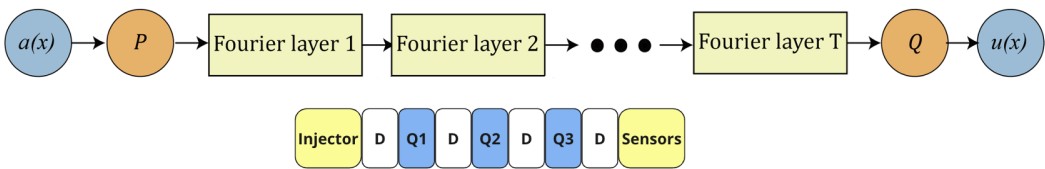

Figure 1: Top: architecture of Fourier Neural Operators, from the work of Li et al. (2020). Bottom: experimental module lattice with an injector, drift zones, quadrupoles and sensors in the target.

On the other hand, DRL techniques consist of two fundamental elements: the environment and the agent. The environment represents the model of the problem to be solved, with states ($S_t$) containing information about the system at time $t$; whereas the agent is an algorithm that uses information obtained from the environment to make decisions (Graesser & Keng, 2019). Following the execution of an action ($A_t$), the environment transitions to the subsequent state ($S_{t+1}$), and a reward function assesses whether the agent is progressing toward or deviating from the objective. These challenges are framed as Markov Decision Processes, where learning occurs through the Bellman equations and the agent's experiences through multiple iterations. The primary objective for the agent is to learn a policy function ($\pi(S_t)$) determining optimal actions based on observed states. This function, essentially the agent's brain, is approximated through neural networks.

## 3 METHODOLOGY AND RESULTS

### 3.1 FNO PREDICTIONS

The data required to train the FNO models was generated with simulations of deuteron bunches[1] travelling through the experimental lattice with varying conditions. The input functions are taken to be the constant parameters controlling the shape of the beam. First, the injected Gaussian distribution of particles ($s_x(z = 0)$, $s_y(z = 0)$) lies within the interval [0.25, 0.75] $mm$. Second, the magnetic strengths of the quadrupoles ($k_1$, $k_2$, $k_3$) range between [-2.5, 2.5] $Tm^{-1}$. The output functions that the model predicts are the beam envelopes (in this case, the standard deviations), $s_x(z)$ and $s_y(z)$, along the lattice longitudinal axis $z$. The end of the line is located at a distance $z = L$.

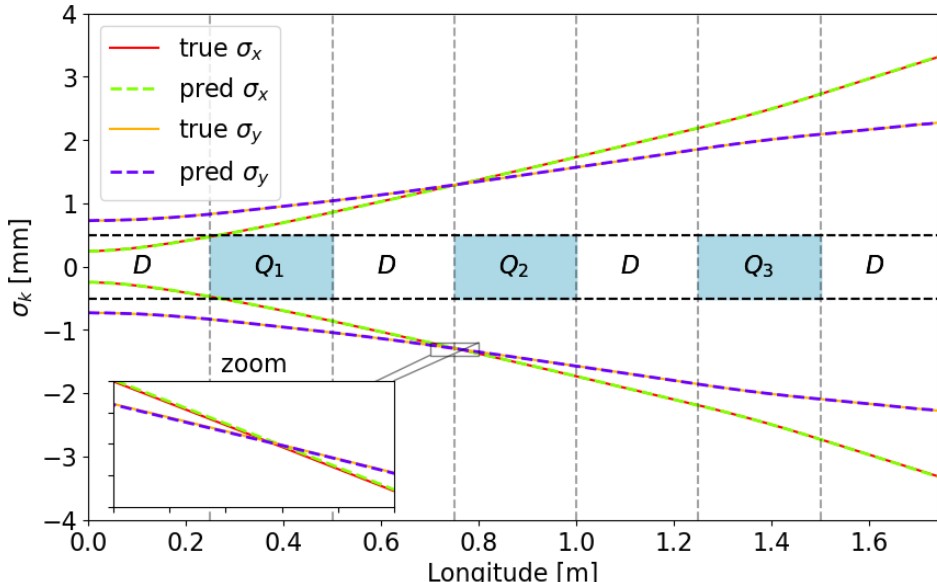

Figure 2: Predicted and true standard deviations, on the $x$ and $y$ axes, of the beam along the longitudinal axis for a test instance. These results were computed with the variable injection FNO model. The residual errors are so small that the predicted values visually overlap the real values (see Fig. 5). The negative component is plotted to display the symmetry of the beam in each axis.

Two models were trained using the nvidia-modulus[2] framework. The first model (with constant particles injection) took 1331 simulations, whereas the second model (with variable injection) made use of 11979 simulations. The envelopes solution space con be found in Appendix G. Prediction results of the variable injection model (for one example on the test set) are displayed in Fig. 2. A summary showcasing the results obtained in the test sets, including speed-up factors in comparison to OPAL simulations and percentage errors, is presented in Tables 1 and 2 of Appendix A. The hyperparameters of the FNO architecture can be seen in Table 3. Other test examples and the residual errors are displayed in Appendix B. The box diagram of the model can be find in Appendix D.

### 3.2 DRL AGENT

The FNO model with variable injection is used to train a DRL agent in order to sequentially reach different beam shapes at the end of the line. This is achieved through the manipulation of quadrupoles acting as actuators. In each step, which corresponds to a deuteron bunch crossing, the agent is able to incrementally modify the 3 quadrupoles ($\Delta k_1$, $\Delta k_2$, $\Delta k_3$) with a set limit[3]. The default state of the quadrupoles is ($k_1 = 0$, $k_2 = 0$, $k_3 = 0$). The observation space for this

---

[1]Each bunch consists of 20000 particles.
[2]https://github.com/NVIDIA/modulus-sym
[3]It is a hyperparameter set to 0.25 $Tm^{-1}$.

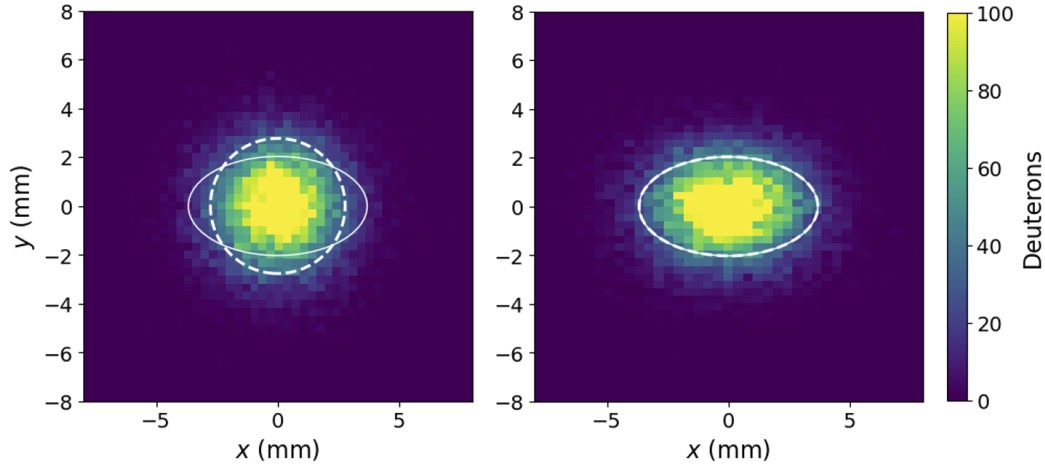

Figure 3: Deuteron distributions observed by the agent (20000 particles). Left: initial configuration. Right: final state after taking 9 actions. Solid line: configuration to reach. Dashed line: current state of the agent. This evaluation corresponds to the first point of Table 6 in Appendix C, where other solutions are shown.

agent is defined with the standard deviations $(s_x(L), s_y(L))$ at the target , the absolute state of the quadrupoles and the target configuration to reach. The reason to limit the observation space of standard deviations to $z = L$ is that in real applications only the distribution at the target would be available using a camera or other sensor. Finally, the reward function was defined as the normalized negative euclidean distance between the target and the current configuration, with a 10 % distance threshold for the winning condition.

The wrapper for the DRL environment was built with the Gymnasium tool(Towers et al., 2023) and then a Proximal Policy Optimization (PPO) (Schulman et al., 2017) agent was trained with the Stable-Baselines 3 framework (Raffin et al., 2021). Results for one example can be seen in Fig. 3, where the agent reaches the desired configuration $(s_x(L) = 3.7, s_y(L) = 2.0)$ within 9 actions. The 2-dimensional deuteron distributions are shown for the initial and final states at the target. Diagrams with the training loop, the environment and the policy network can be seen in Appendix E.Tables with the hyperparameters and the observations and actions spaces are shown in Appendix C. Additionally, Table 6 displays the agent evaluation and performance on other targets.

### 3.3 ACCELERATOR OPTIMIZATION

Another application of the FNO model comes from the fact that it is a differentiable PyTorch model, thus the gradients can be utilized in inverse problems to find the optimal input parameters for a given output (Zhou et al., 2024). In this case, we built a loss function with the target parameters of the beam at the end of the lattice and used Stochastic Gradient Descent (SGD) to obtain the optimal values of the quadrupoles and particles injections.

We applied the SDG optimization to both constant and variable injection models, with euclidean distance to the target parameters as loss function, and a early stopping tolerance of $0.005$, constraining the inputs to the domain in which the FNO has been trained. This method lead with both FNO models to the optimal design in less than 32 seconds, a fast process compared with other techniques, such as evolutionary algorithms or DRL methods. Optimization loop diagram can be found in Appendix D and tables with results for different runs and the parameters used by the optimizer are shown in Appendix F.

# 4 CONCLUSIONS AND OUTLOOK

The trained FNO models exhibit a remarkable capability to predict beam envelopes at a speed 3 orders of magnitude faster than traditional simulations with OPAL, achieving a maximum percentage error below 2 %. This accelerated prediction not only facilitates rapid inference of beam properties but also enables the training of DRL agents within feasible time frames. The DRL agent achieves the optimal policy in less than 20 minutes, a substantial improvement compared to the estimated 225 hours required with OPAL for 270k steps. Additionally, the differentiable nature of the FNO model is leveraged to develop an optimizer for identifying optimal accelerator parameters, leading to significantly reduced optimization times—merely seconds compared to alternative methods such as DRL optimization or evolutionary algorithms.

These results underscore the synergistic potential of combining DLSMs with other Deep Learning techniques, presenting promising avenues for advancing control and optimization strategies not only in the IFMIF-DONES accelerator but also in other complex scientific facilities. In the context of nuclear fusion power plant development, this study highlights the promising role of AI techniques in advancing fusion research, as can also be seen in the work of Degrave et al. (2022). Finally, the next step of this research will be to apply these techniques to each of the modules of the IFMIF-DONES facility.

ACKNOWLEDGMENTS

We want to express our sincere gratitude to CDTI spanish institution for the DONES-FLUX project funding. Additionally, we wish to acknowledge and thank the IFMIF-DONES consortium as well as the university of Granada for their role played in the development of this work.

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

# A APPENDIX: FNO MODEL SUMMARY

Table 1: Percentage errors for both FNO models and the two predicted functions. Top: constant injection. Bottom: variable injection.

| Error | $s_x(z)$ | $s_y(z)$ |
|---|---|---|
| Max absolute % error | 0.80 | 0.69 |
| Mean absolute max % error | 0.31 | 0.25 |
| Mean absolute % error | 0.08 | 0.08 |
| Mean % error | 0.03 | 0.01 |
| Max absolute % error | 1.90 | 1.57 |
| Mean absolute max % error | 0.74 | 0.65 |
| Mean absolute % error | 0.33 | 0.28 |
| Mean % error | 0.07 | -0.05 |

Table 2: Inference times and speed up factors for both FNO models. Single stands for only one simulation or inference, whereas multiple means the inference was performed on a set of instances.

| Model | Single [ms] | Multiple [ms] | Factor (single) | Factor (multiple) |
|---|---|---|---|---|
| Constant injection | 15.3 | 55.4 (267 instances) | $2 \times 10^2$ | $1.5 \times 10^3$ |
| Variable injection | 15.5 | 370 (2396 instances) | $2 \times 10^2$ | $2 \times 10^3$ |

Table 3: Nvidia-modulus hyperparameters for the FNO architecture.

| Hyperparameter | Value |
|---|---|
| scheduler | tf_exponential_lr |
| optimizer | adam |
| loss | sum |
| decoder.nr_layers | 1 |
| decoder.layer_size | 32 |
| fno.dimension | 1 |
| fno.nr_fno_layers | 4 |
| fno.fno_modes | 12 |
| scheduler.decay_rate | 0.95 |
| scheduler.decay_steps | 1000 |
| training.rec_results_freq | 500 |
| training.max_steps | 10000 |
| batch_size.grid | 32 |
| batch_size.validation | 32 |

## B    APPENDIX: OTHER FNO TEST EXAMPLES

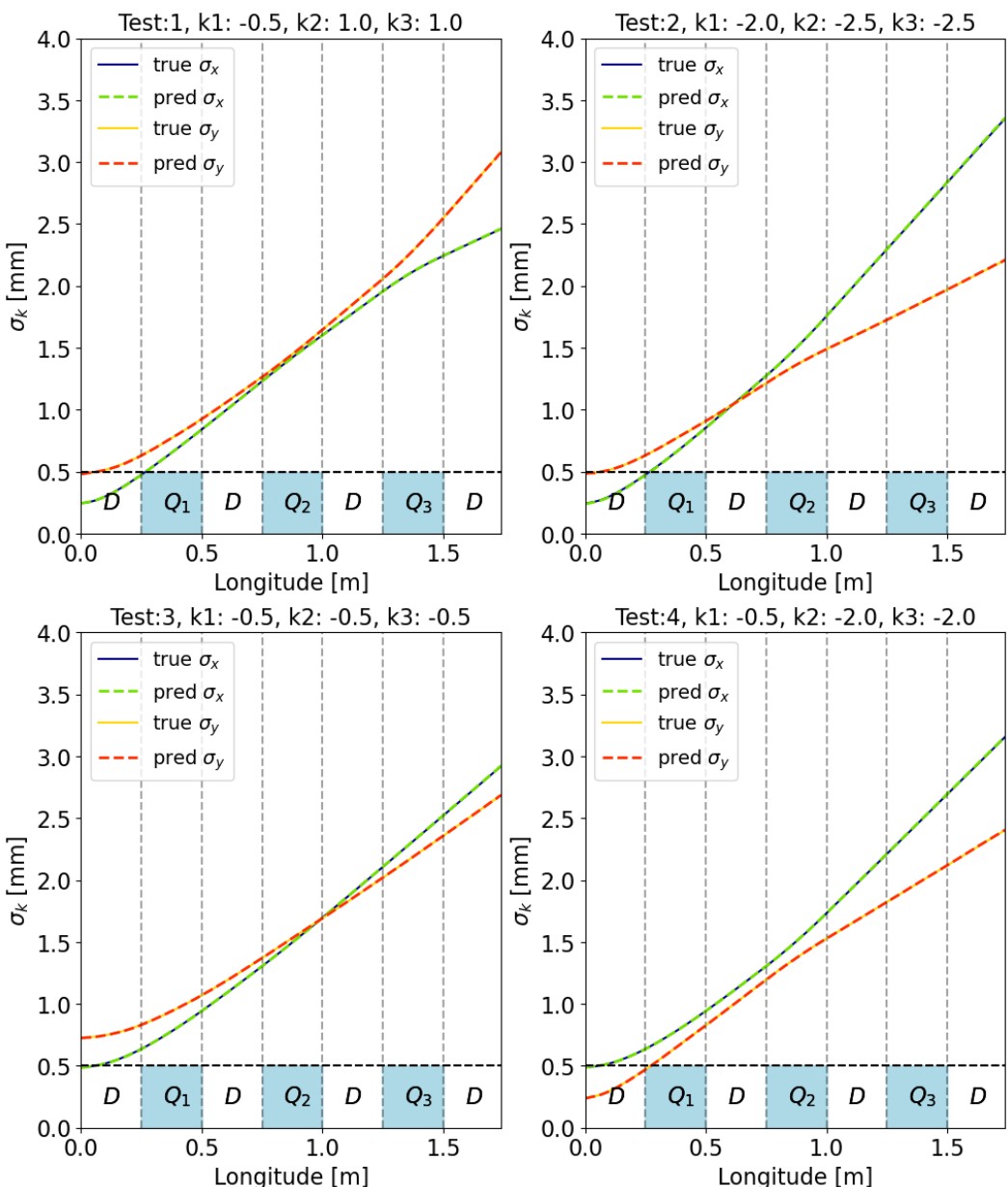

Figure 4: Beam envelopes predicted by the FNO model with variable injection for 4 examples in the test set. Each instance represents different injection and quadrupole values. The errors, as can be seen in Fig. 5, are so small that the predicted and real curves completely overlap.

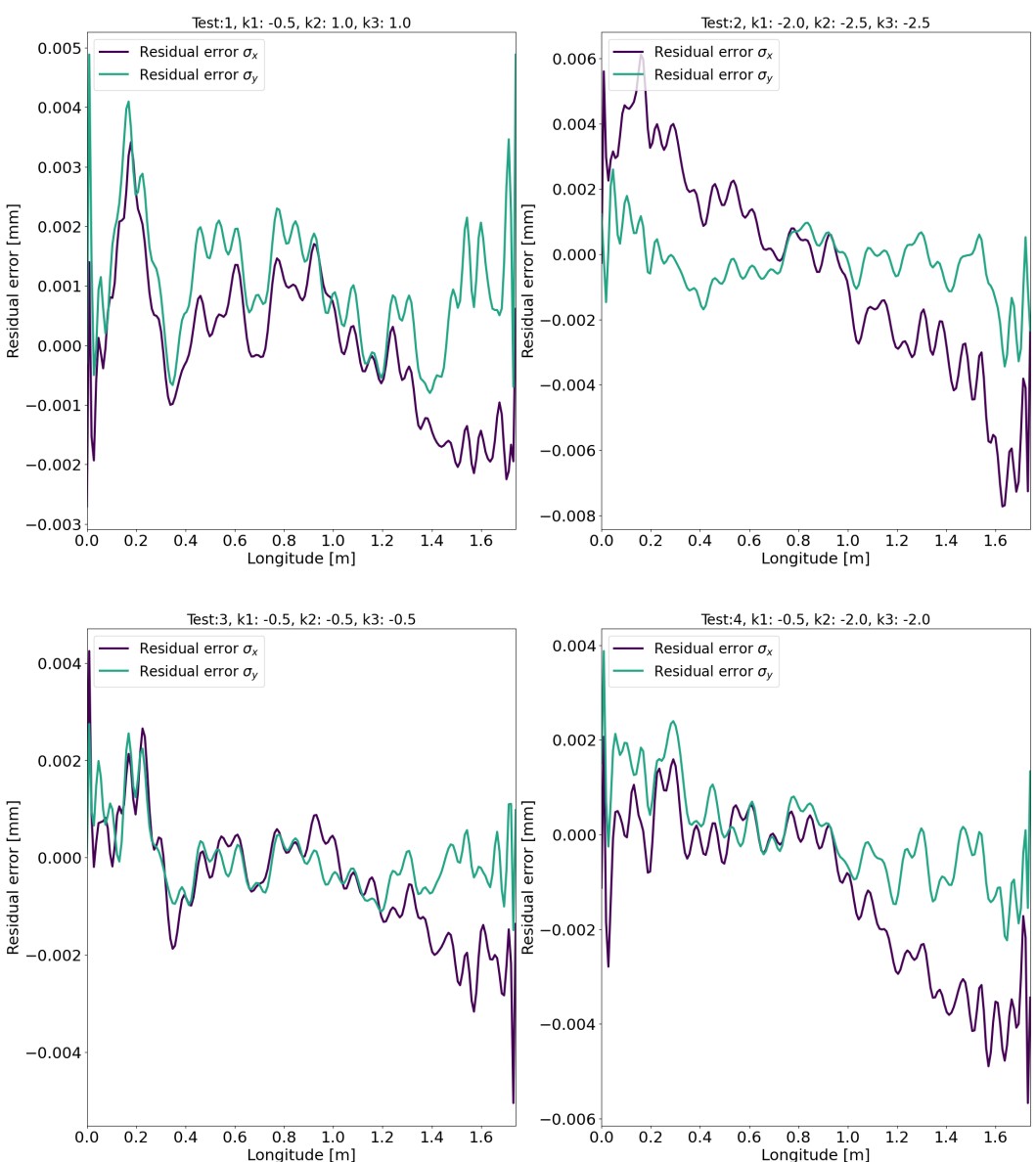

Figure 5: Beam envelope residual errors for the same test examples from Fig. 4. All the errors are smaller than $6 \times 10^{-3}$ mm.

## C  APPENDIX: DRL AGENT SUMMARY

Table 4: PPO hyperparameters used with Stable-Baselines 3. The agent training was stopped at 270k steps.

| Hyperparameter | Value |
|---|---|
| policy | MultiInputPolicy |
| learning_rate | 0.0008 |
| n_steps | 2048 |
| batch_size | 64 |
| n_epochs | 10 |
| gamma | 0.902 |
| gae_lambda | 0.95 |
| clip_range | 0.2 |
| clip_range_vf | null |
| normalize_advantage | true |
| ent_coef | 0 |
| vf_coef | 0.5 |
| max_grad_norm | 0.5 |
| total_timesteps | 1000000 |
| act_function | tanh |
| layers | [64, 64] |

Table 5: Spaces for the DRL problem and the gymnasium wrapper, with the minimum and maximum values per variable. Top: observations space. Bottom: actions space.

| Variable | Min | Max |
|---|---|---|
| $k_j, (Tm^{-1})$ | -2.5 | 2.5 |
| $s_x \ (mm)$ | 0 | 4 |
| $s_y \ (mm)$ | 0 | 4 |
| Target $s_x \ (mm)$ | 0 | 4 |
| Target $s_y \ (mm)$ | 0 | 4 |
| $\Delta k_j \times 4 \ (Tm^{-1})$ | -1 | 1 |

Table 6: Agent evaluation for different configurations. It includes the target configuration, the solution reached by the agent, the number of actions taken, the normalized distance between both configurations (has to be smaller than the 0.1 threshold value for the victory condition) and the quadrupole values. The first element is the one displayed in Fig. 3.

| Target $(s_x, s_y)$ | Solution $(s_x, s_y)$ | Actions | Distance | $k_1$ | $k_2$ | $k_3$ |
|---|---|---|---|---|---|---|
| (3.70, 2.00) | (3.68, 2.03) | 9 | 0.03 | -2.25 | -2.25 | -2.25 |
| (3.24, 2.34) | (3.26, 2.35) | 5 | 0.04 | -1.22 | -1.25 | -1.25 |
| (3.08, 2.50) | (3.06, 2.51) | 3 | 0.06 | -0.72 | -0.75 | -0.74 |
| (2.65, 2.91) | (2.71, 2.84) | 2 | 0.09 | 0.50 | 0.16 | 0.45 |
| (2.90, 2.65) | (2.89, 2.67) | 2 | 0.09 | 0.03 | -0.50 | -0.35 |
| (2.55, 3.08) | (2.52, 3.05) | 3 | 0.09 | 0.75 | 0.73 | 0.75 |
| (2.25, 3.38) | (2.27, 3.35) | 6 | 0.04 | 1.50 | 1.50 | 1.50 |
| (1.95, 3.79) | (1.97, 3.77) | 10 | 0.02 | 2.50 | 2.50 | 2.50 |

## D APPENDIX: FNO AND OPTIMIZATION DIAGRAMS

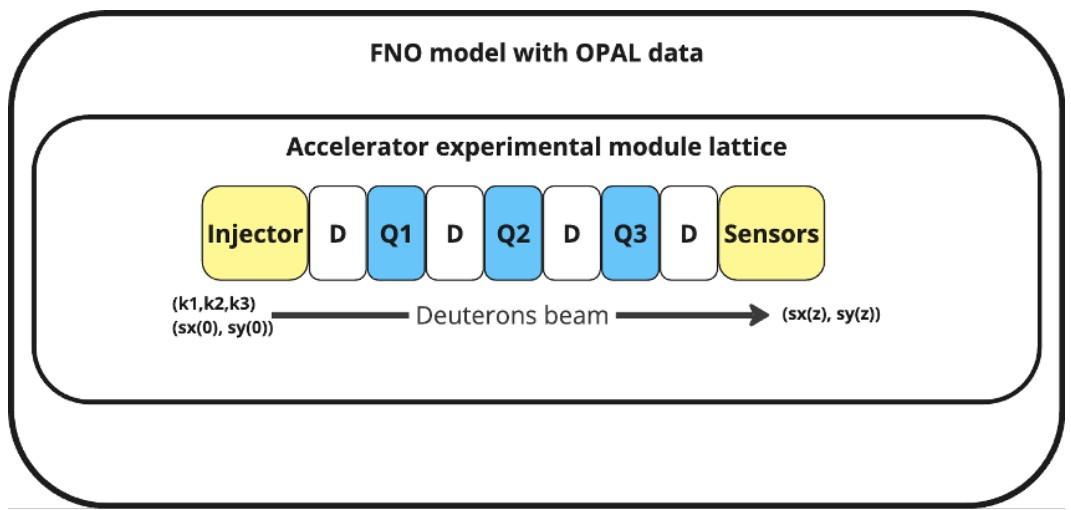

Figure 6: FNO diagram, it takes the magnetic forces and initial standard deviation of particles distribution as input and outputs the standard deviation particle distribution along the lattice.

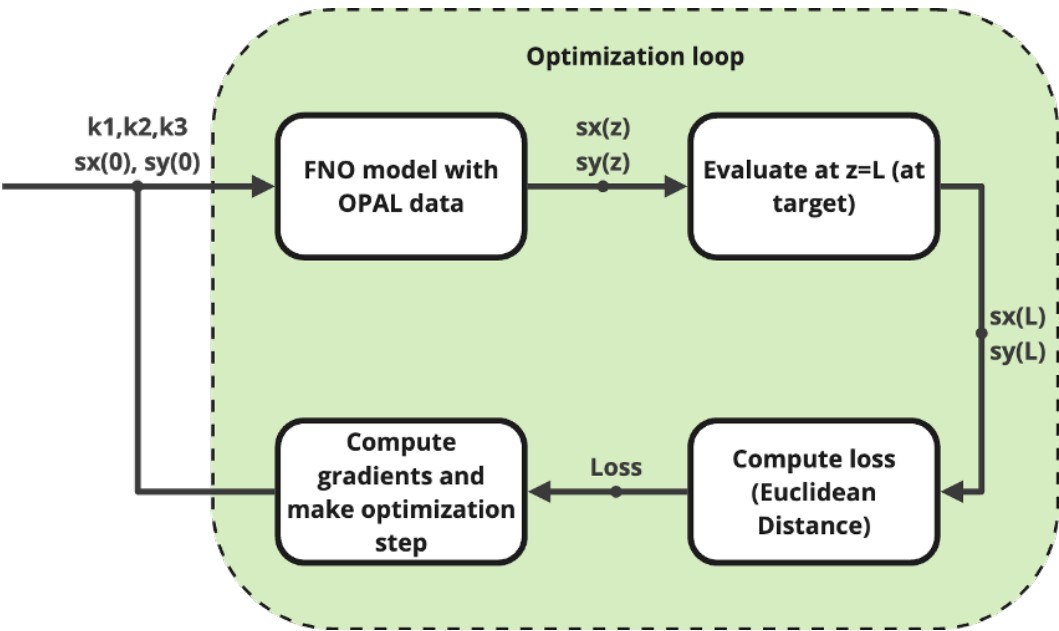

Figure 7: Optimization loop diagram, it searches the optimal magnetic forces and initial standard deviation of particles distribution that give a objective distribution at the target. The loss function is defined as the euclidean distance of the actual and objective particle distribution at the target

# E  APPENDIX: DRL DIAGRAMS

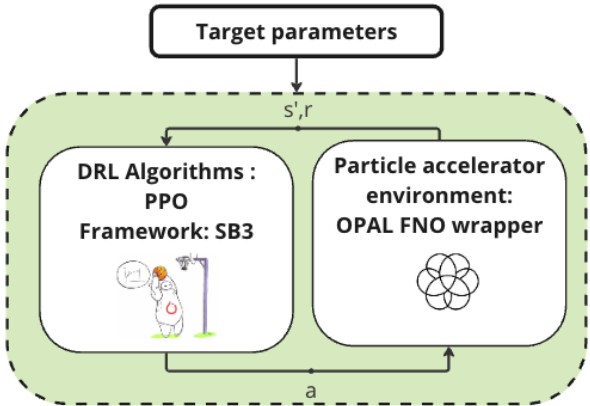

Figure 8: Training loop of the DRL agent. In each iteration, it performs an action $a$, obtains a reward $r$ and the environment advances to the next state $s'$.

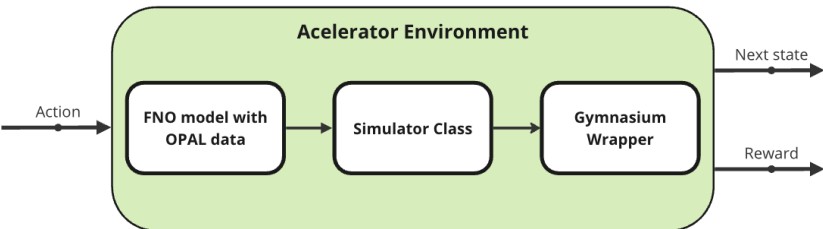

Figure 9: Environment used for training the DRL agent.

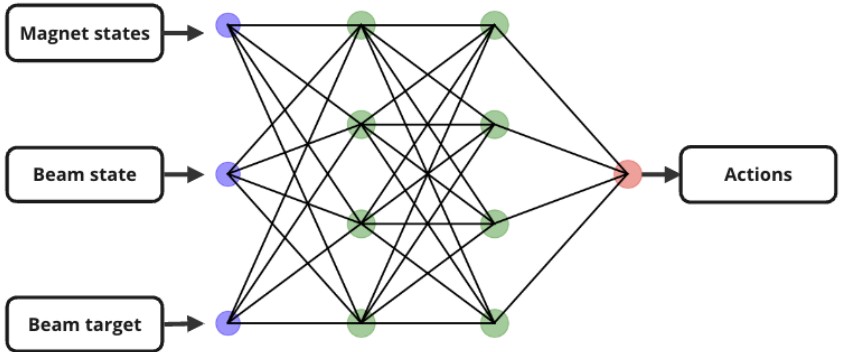

Figure 10: Trained policy. Input nodes (7): the quadrupole absolute states, the beam current state at the target and the configuration to reach. Output nodes (3): an action per quadrupole. The hidden layers are 2x64.

## F  APPENDIX: DESIGN OPTIMIZATION WITH THE FNO MODEL

Table 7: Design optimization with FNO for different configurations. It includes the target configuration, the solution reached by the SGD optimizer, the distance between both configurations (has to be smaller than the 0.005 threshold value for the early stopping condition) and the quadrupole values.

| Target $(s_x, s_y)$ | Solution $(s_x, s_y)$ | Loss (Distance) | time [s] |
|---|---|---|---|
| (3.70, 2.00) | (3.700, 1.995) | 0.0019 | 3.9 |
| (3.24, 2.34) | (3.237, 2.342) | 0.0029 | 0.2 |
| (3.08, 2.50) | (3.083, 2.497) | 0.0044 | 26.0 |
| (2.65, 2.91) | (2.652, 2.906) | 0.0041 | 6.1 |
| (2.90, 2.65) | (2.901, 2.650) | 0.0042 | 2.4 |
| (2.55, 3.08) | (2.546, 3.086) | 0.0035 | 11.2 |
| (2.25, 3.38) | (2.254, 3.377) | 0.0047 | 2.9 |
| (1.95, 3.79) | (1.969, 3.793) | 0.0049 | 32.0 |

Table 8: Optimization hyperparameters using the FNO model

| Hyperparameter | Value |
|---|---|
| scheduler | exponential_lr |
| optimizer | SDG |
| loss | Euclidean distance |
| scheduler.gamma | 0.999 |
| max_steps | 100000 |
| early_stop_loss | 0.005 |
| learning_rate | 1 |

## G  APPENDIX: ENVELOPES SOLUTION SPACE

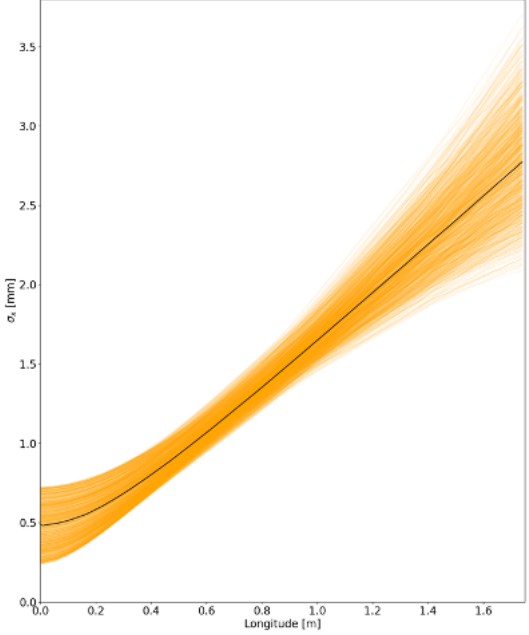

Figure 11: Envelopes solution space generated using random input parameters.

