# OpenReview forum: "Applications of Fourier Neural Operators in the Ifmif-Dones Accelerator"
_ICLR.cc/2024/Workshop/AI4DiffEqtnsInSci — AI4DiffEqtnsInSci @ ICLR 2024 Poster_

### Official Review · Reviewer_S5U3 · 2024-02-23
**Review of “Applications of Fourier Neural Operators in the IFMIF-DONES Accelerator”.**

**Rating:** 9
**Confidence:** 4

**Review:**

This uses data driven methods to help with shaping the beam envelopes in the IFMIF-DONES linear accelerator. The authors do this in two steps. They first train a neural operator to predict the beam envelope given the tuning parameters. They then show that this low-cost map between tuning choice and outcome can be used to cost effectively train a control agent to choose tuning parameters given the desired envelope. The authors also show that the neural operator map can also be directly optimized over to produce the desired tuning parameters. Outside of the contribution to the IMFIF-DONES workflow, the paper introduces a novel workflow for reinforcement learning by first training a low-cost model to replace a numerical integrator and then using it to perform the large number of iterations necessary for reinforcement learning.

My knowledge on the problem domain of nuclear fusion is minimal, but the paper is well written and demonstrates clear success with the implementation proposed. Additionally, the pipeline used to train the model may be useful in other applications where low-cost replacements for numerical integrators may allow reinforcement learning techniques to be more easily applied. My only hesitation is my lack of familiarity with the challenge and scope of the physical system being studied or existing best practices in the field.

Minor Comments:
1)	In section 2 you reference figure 6 where I think you intend to reference figure 1.
2)	In the second paragraph of section 3.3 “both constant an variable” should be “both constant and variable”
3)	As someone outside of the field, understanding the structure of the data being explored is a bit of a challenge. It may be helpful to provide additional visualizations of the range of envelopes in the appendix.
4)	There is no baseline method provided, it would be helpful to discuss how the tuning parameters are selected in the current state-of-the-art.

---

### Official Review · Reviewer_F7V3 · 2024-02-27
**Review of FNO paper**

**Rating:** 7
**Confidence:** 4

**Review:**

This is a review of the manuscript, "Applications of Fourier neural operators in the IFMIF-DONES accelerator," submitted to the ICLR 2024 Workshop on AI4DifferentialEquations In Science. The paper describes a Fourier neural operator (FNO) trained to predict the envelope parameters of neutron beams in the IFMIF-DONES facility. Further, a deep reinforcement learning (DRL) model is trained for optimization and/or control purposes. My review is summarized below in point form.

1. The FNO provides a nice speed-up, but it is also (presumably) trained on $\{s_x, s_y\}$ data as a function of $z$. Are $\{s_x, s_y\}$ data actually available as a (near-)continuous function of $z$? Or are individual particle trajectories somehow measured? What is the expected fidelity of such data, and how does the FNO's performance deteriorate in the context of realistic uncertainties? For instance, does parameterizing the beam envelope in terms of the standard deviation of the particle distribution lead to potential model errors? The authors should provide a more comprehensive phenomenological description of the physical system. They should answer such questions, and better motivate their chosen surrogate model form. Doing so will help situate readers from other fields.

2. Did the authors consider alternative operator network architectures? In many cases, a simple multilayer perceptron (MLP) will suffice. Especially in the present context, which features low-dimensional input and output spaces and a limited number of control parameters. A simpler network might perform equally well! The authors should test an MLP of similar depth, having the same number of trainable parameters as their FNOs, to assess whether the FNO configuration is beneficial, per se.

3. The authors use stochastic gradient descent to optimize $\{s_x(0), s_y(0), k_1, k_2, k_3\}$ to achieve a target output envelope, $\{s_x(L), s_y(L)\}$. As presented, this seems like a very simple optimization problem that could be solved using vanilla gradient descent. Why is it necessary to use the stochastic version?

4. Related to the first and third points, the need for a DRL controller is unclear. Is the controller required for real-time applications? And if $\{s_x, s_y\}$ data are available as a function of $z$ to train the model (i.e., in the absence of ``comprehensive knowledge regarding the [governing equations]''), why does the controller only have access to the data at $z = L$? Wouldn't it be feasible to reduce the number of requisite actions by feeding the controller a richer observation?

5. The authors should comment on the performance of their FNO outside the training envelope. (It can be difficult to operationalize the notion of extrapolation in many deep learning settings, of course, but the present demonstration is relatively low-dimensional.)

---

### Meta-Review · Area_Chair_Got5 · 2024-02-28

**Recommendation:** Accept (Poster)

**Metareview:**

Dear Authors,

Thank you for submitting the draft.

Both reviewers agree that the work contains an interesting contribution. It is expected that authors will be addressing comments by the reviewers in the final draft.

regards

AC

---

### Decision · Program_Chairs · 2024-02-29

Accept (Poster)